# A Smart Textile Band Achieves High-Quality Electrocardiograms in Unrestrained Horses

**DOI:** 10.3390/ani12233254

**Published:** 2022-11-23

**Authors:** Persephone McCrae, Hannah Spong, Ashley-Ann Rutherford, Vern Osborne, Amin Mahnam, Wendy Pearson

**Affiliations:** 1Department of Animal Biosciences, University of Guelph, Guelph, ON N1G 2W1, Canada; 2Department of Research and Development, Myant Inc., Toronto, ON M9W 1B6, Canada

**Keywords:** equine, cardiology, electrocardiogram, smart textile, textile computing

## Abstract

**Simple Summary:**

Electrocardiography is a method used to understand equine cardiovascular health and fitness. While electrocardiograms (ECGs) are essential, the devices typically utilized are limited by their reliance on adhesive electrodes. Therefore, the aim of this study was to compare ECG quality obtained using a smart textile band to the standard adhesive electrodes. ECGs were recorded simultaneously using both electrode types in stalled horses. We did not observe any significant differences in ECG signal quality, the degree of motion artifacts, or the output of heart rate. These results indicate that smart textile electrodes are a reliable alternative to adhesive electrodes for ECG data collection in horses at rest.

**Abstract:**

Electrocardiography (ECG) is an essential tool in assessing equine health and fitness. However, standard ECG devices are expensive and rely on the use of adhesive electrodes, which may become detached and are associated with reduced ECG quality over time. Smart textile electrodes composed of stainless-steel fibers have previously been shown to be a suitable alternative in horses at rest and during exercise. The objective of this study was to compare ECG quality using a smart textile girth band knit with silver and carbon yarns to standard adhesive silver/silver chloride (Ag/AgCl) electrodes. Simultaneous three-lead ECGs were recorded using a smart textile band and Ag/AgCl electrodes in 22 healthy, mixed-breed horses that were unrestrained in stalls. ECGs were compared using the following quality metrics: Kurtosis (k) value, Kurtosis signal quality index (kSQI), percentage of motion artifacts (%MA), peak signal amplitude, and heart rate (HR). Two-way ANOVA with Tukey’s multiple comparison tests was conducted to compare each metric. No significant differences were found in any of the assessed metrics between the smart textile band and Ag/AgCl electrodes, with the exception of peak amplitude. Kurtosis and kSQI values were excellent for both methods (textile mean k = 21.8 ± 6.1, median kSQI = 0.98 [0.92–1.0]; Ag/AgCl k = 21.2 ± 7.6, kSQI = 0.99 [0.97–1.0]) with <0.5% (<1 min) of the recording being corrupted by MAs for both. This study demonstrates that smart textiles are a practical and reliable alternative to the standard electrodes typically used in ECG monitoring of horses.

## 1. Introduction

Horses have a higher incidence of cardiac arrhythmias at rest than any other domestic species [1,2]. Arrhythmias at rest and/or during exercise may be clinically irrelevant or may be associated with poor performance and even sudden cardiac death (SCD) [3,4,5,6,7,8]. SCD during equestrian sports is defined as the acute collapse and death in an otherwise healthy horse during, or immediately after, exercise [7]. The occurrence of SCD is ten times more common in horses than in humans [7,9]. Arrhythmias have been associated with SCD, and cardiovascular disease is thought to be responsible for 14% and 24% of sudden death cases and episodes of collapse, respectively [7,10,11,12]. However, arrhythmias in horses are not always associated with clinical signs which can alert the owner, and simple auscultation is not always reliable. Therefore, electrocardiograms (ECGs) are needed to make a definitive diagnosis [13]. 

Conventional ECG devices are typically only employed in a clinical or research setting, limiting their use. These devices are often expensive, cumbersome, and require training. Smart textiles have gained popularity in the monitoring of human health due to their accessibility, reliability, and comfort [14,15,16]. They are also increasingly being used within veterinary medicine, particularly due to their perceived ability to promote animal comfort and reduce stress [17]. One application of smart textiles is ECG monitoring using non-adhesive, flexible electrodes. Electrodes and traces for ECG monitoring are developed either by surface printing conductive materials on fabric or utilizing conductive yarns woven directly into the textile [18]. 

Distortion of ECGs by motion artifacts (MAs) is a common problem that limits analysis and can lead to misinterpretation [2,19]. MAs are particularly problematic during monitoring of animals due to their unpredictable movements. In particular, restraint of horses during monitoring has been shown to cause a stress response which may result in an unrepresentative ECG [20]. It has previously been shown that ECG data collected from horses at rest with textile electrodes can have fewer MAs than data collected with standard silver/silver chloride (Ag/AgCl) electrodes [21]. However, the authors still observed that MAs corrupted approximately 35% of the data collected with the textile electrodes over a 60 min period [21]. In a more recent study, Felici et al. found that ECG signal quality was the same between textile and Ag/AgCl electrodes during standardized exercise tests at a walk, trot, and gallop on a treadmill, though it was variable across gait and time [22]. Both studies utilized textile electrodes knit with stainless steel fibers, which, while durable, are challenging to knit in scale due to their size and stiffness, possibly leading to skin irritation. Moreover, the prevalence of MAs when using stainless steel has previously been reported in humans [23,24,25]. This is because stainless steel produces a polarizable electrode [26], where a majority of the charge necessary for biopotential measurement crosses the electrode-electrolyte interface capacitively instead of via reactions. This results in the production of overpotentials which are recorded as noise in addition to the recorded biopotential. By comparison, Ag/AgCl electrodes, which are considered nonpolarizable, are the standard for ECG recording, as they exhibit significantly less low-frequency electric noise. 

In this study, we investigated a fully knit textile band with electrodes consisting of carbon and silver-plated yarns. Due to the nylon core of these yarns, the fibers are flexible, easy to knit, and can conform to the curvature of the body. Therefore, the objective of this study was to compare ECG signal quality obtained using textile electrodes composed of silver and carbon yarns to Ag/AgCl electrodes in horses at rest.

## 2. Materials and Methods

### 2.1. Animals and Housing Conditions

In total, 22 healthy, mixed-breed horses were recruited for the purpose of this study (mean age 13 ± 5.4 years; 13 mares, 9 geldings; 10 Standardbreds, 9 Warmblood/Warmblood crosses, 1 Quarter horse, 1 Arabian, and 1 Thoroughbred). Horses did not have a history of poor performance or cardiac abnormalities. The study was performed in accordance with the University of Guelph Animal Utilization Protocol (reference No. 4705). All horses were housed separately in standard box stalls (12 × 12 ft). Horses were not restrained during data collection and were permitted to move freely around the stall. Horses were provided *ad libitum* access to hay and water throughout the data collection period. No changes were made to the standard management of any of the horses.

### 2.2. Experimental Protocol

ECGs were simultaneously collected using textile electrodes integrated into a smart textile girth band (Myant Inc., Skiin Equine; Figure 1) and Ag/AgCl electrodes (SKINTACT W-601). The textile girth band was composed of a lycra/polyester blend. An adjustable closure on the band allowed for customization of fit. Bands ranged between 165 and 180 cm in length and were found to fit all horses used in the study. Textile electrodes were 4 cm squares composed of silver- and carbon-coated yarns. Ag/AgCl electrodes were adhered to the horses’ skin directly above the textile electrodes. The skin was not prepared, and the hair was not shaved for the application of either electrode type. A total of five electrodes of each type (textile and Ag/AgCl) were applied to the horse (caudal to the left and right olecranon, left and right aspects of thoracic vertebrae 8, with the ground placed on the right side of the body) to construct a three-lead ECG trace (Figure 2). A salt- and chloride-free electrically conductive gel (Parker, Spectra 360 Electrode Gel) was applied to both electrode types. The Ag/AgCl electrodes already contained a solid adhesive gel of 2 cm^2^; however, an additional small amount of the conductive gel was applied to increase the moisture content. Silver-coated yarns were knit into the textile band to connect the textile electrodes to the recording device, while cable wires were snapped directly to the button of the Ag/AgCl electrodes. 

Identical devices (Myant Inc., Skiin Monitoring System) were used to record ECGs from both types of electrodes at a sampling frequency of 320 Hz. Data were transmitted via a Bluetooth connection to two mobile phones. Recordings lasted one hour in duration. Data were viewed in real time to confirm quality among the three channels and were later uploaded to a computer for analysis.

### 2.3. Data Analysis

Data were processed using a custom Python script. Baseline wander was removed using a high-pass Butterworth filter with a cut-off frequency of 0.5 Hz, using a forward-backward technique for a zero-phase response. To reduce high-frequency noise, a low-pass 4th-order Butterworth filter with a cut-off frequency of 35 Hz was used. Power-line noise was removed using a 2nd-order Butterworth band-stop filter with a frequency of 58–62 Hz [22,27,28,29]. 

ECG data were windowed in 10 s segments, mutually overlapping by 5 s. ECG quality of the segmented signals was compared using the Kurtosis (*k*) value, Kurtosis Signal Quality Index (kSQI), percentage of motion artifacts (%MA), peak signal amplitude, and heart rate (HR). *K* value is considered to be the single best metric for assessing ECG signal quality, where the distribution of peaks is compared to a Gaussian distribution [27]. For each segment and for each lead, the empirical estimate of *k* of the discrete signal $x_{i}$ of that lead was calculated using the following equation, as has previously been described [22,27].
$$\hat{K}=\frac{1}{M\;}\;\overset{M}{\underset{i=1}{\sum }}{\left[\frac{x_{i}-{\hat{\mu }}_{x}}{\hat{\sigma }}\right]}^{4}$$
where ${\hat{\mu }}_{x}$ and $\hat{\sigma }$ are the empirical estimate of the mean and standard deviation of $x_{i}$, respectively, and *M* is the number of samples. As previously suggested, a *k* value greater than 5 indicates good quality ECG signal, while a *k* value lower than 5 indicates that the signal is impacted by motion artifacts or interference [22,27]. The kSQI was calculated from the number of windows where the *k* value was greater than 5, divided by the total number of windows, as previously described [22,28,30]. The percentage of each recording distorted by motion artifacts (%MA) was determined by manually counting the duration of segments of the file that were not of diagnostic quality where P waves and/or QRS complexes were not identified [31,32]. Peak signal amplitude was calculated as the amplitude of the R peak to S peak. HR was calculated from R peaks. The root mean square error (RMSE) was calculated for each channel to find the degree of error between the two electrode types in HR outputs. 

Sample size calculations were performed using standard deviations, confidence level (0.05), and power (0.80) of kSQI from textile and Ag/AgCl electrodes collected during a pilot test conducted on five horses. Data from each electrode type were averaged between the three channels for a single value. These calculations revealed a required sample size of 6 horses. The assumptions of normality and equal variance were assessed using the Shapiro–Wilk normality tests. All normally distributed data are presented as mean ± standard deviation, and all non-normal data are presented as median and interquartile range. Two-way ANOVA with Tukey’s multiple comparison tests was performed to compare electrode types. All statistical tests were performed using GraphPad Prism (9.1.0) with statistical significance set at *p* ≤ 0.05.

## 3. Results

Twenty-two pairs of simultaneous three-lead ECG tracings were obtained using textile and Ag/AgCl electrodes (two recordings per horse, one with each electrode type). Both electrode types were consistently capable of producing high-quality ECG signals across the three channels (Figure 3). No horses had to be excluded from the study, and all tolerated the textile girth band well. Approximately half of the horses had previously worn the girth band, while the remainder of the horses were naïve to it. No behavioral changes were observed in any of the horses throughout data collection. The band was not observed to cause any discomfort, and there were no indications of tenderness, rubbing, or hair loss. Furthermore, the horses also did not attempt to damage, chew, or remove the bands, indicating that it can be utilized for prolonged monitoring. 

### 3.1. Kurtosis and kSQI

No differences were observed between textile and Ag/AgCl electrodes for both k and kSQI values (Figure 4). The k values were above the optimal value of 5, with a mean of all three channels of 21.8 ± 6.1 and 21.2 ± 7.6 for textile and Ag/AgCl electrodes, respectively. The median kSQI values across all channels were close to the maximum value of 1 (textile: 0.98 [0.92–1.0], Ag/AgCl: 0.99 [0.97–1.0]).

### 3.2. Percentage of Motion Artifacts

The %MA was low for both electrode types, with no differences found between the two (Figure 5). Median %MA from the three channels were 0.35% (0.03–0.88) (~13 s) for textile electrodes and 0.09% (0.11–0.79) (~3 s) for Ag/AgCl electrodes.

### 3.3. Peak Amplitude

Peak amplitude was greater in data collected with textile electrodes for channels I and III (*p* = 0.006, *p* = 0.012, respectively), with no differences observed for channel II (*p* = 0.08) (Figure 6). Mean peak amplitudes for all three channels were 1.74 ± 0.48 mV and 1.46 ± 0.42 mV for textile and Ag/AgCl electrodes, respectively. 

### 3.4. Heart Rate

The calculated HR was within normal resting limits for all horses for both electrode types. HR output was found to be the same for both electrode types, with mean HRs of 40.61 ± 5.38 and 40.58 ± 5.41 for textile and Ag/AgCl, respectively (Figure 7). The root mean square error was small for all channels (ch. I: 0.85 bpm [0.52–2.0], ch. II: 1.18 bpm [0.57–3.4], ch. III: 0.72 bpm [0.54–2.92]).

## 4. Discussion

This study demonstrates for the first time that excellent quality three-lead ECGs can be obtained using a fully integrated smart textile band in horses. No significant differences were observed between the two electrode types for any of the metrics assessed, with the exception of peak signal amplitude. Peak signal amplitude was greater when obtained with textile electrodes for channels I and III (*p* = 0.006, *p* = 0.012, respectively), indicating better contact between the skin and electrode. This may be due to textile settling, where the textile is better able to conform to the body than a rigid Ag/AgCl sensor. Another potential explanation is that textile electrodes retain moisture to maintain humidity between the skin and electrode [33], while Ag/AgCl electrodes dry over time, resulting in deterioration of signal quality [34].

Kurtosis is considered to be the single best indicator of ECG quality, where a low k value (k < 5) is indicative of baseline wander and power-line interference [27,35,36]. Kurtosis and kSQI values have previously been calculated in horses during treadmill exercise (walk, trot, and gallop) using textile electrodes made with stainless steel fibers and Ag/AgCl electrodes (applied with and without glue to shaved skin) [22]. Felici et al. found that a majority k values were above five for the textile electrodes and below five for the Ag/AgCl electrodes [22]. In the present study, average k values of 21.8 and 21.2 were obtained with the textile and Ag/AgCl electrodes, respectively, with no significant differences appreciated between the two. In fact, no k values were less than five for any of the horses, with the lowest k values being 8.6 (textile, channel III) and 5.0 (Ag/AgCl, channel III) on two different horses. Felici et al. also calculated average textile kSQI values of approximately 0.8 to 0.9 during walking at 1.7 m/s, with kSQI values decreasing as speed increased to a trot and gallop. When horses were walked at the same speed at the end of the treadmill test, kSQI values fell to approximately 0.3–0.4. The authors did not find any differences in kSQI values between textile and Ag/AgCl electrodes, except for when glue was used to adhere the Ag/AgCl electrodes, thereby reducing signal quality [22]. While we did not specifically collect data during walking, horses were able to freely move within their stalls; therefore, our data include some degree of walking. We did not observe any differences for kSQI between the two electrode types, which is consistent with Felici et al. for unglued Ag/AgCl electrodes. However, we did observe greater kSQI values for both textile (0.98) and Ag/AgCl (0.99) than had previously been reported [22], indicating that the ECG data reported here were less noisy. These differences in kSQI may be explained by the different electrode materials utilized. The work conducted by Felici et al. utilized electrodes composed of stainless steel, which is associated with increased signal noise due to the polarizable nature of stainless steel [26]. In comparison, the present study utilized textile electrodes that consist of silver-plated and carbon yarns. In nonpolarizable electrodes, there is no production of overpotentials because the current freely passes across the electrode-electrolyte interface. Nonpolarizable electrodes are better suited to ECG recording because they are less prone to artifacts and, therefore, are capable of measuring low-voltage and low-frequency signals, such as those observed in ECGs. These differences in electrode material may also explain the significantly lower degree of MAs observed in this study compared to previous reports. MAs are commonly caused by skin stretching [37] and body movements [38,39,40]. Identification of MAs is typically conducted manually via visual analysis of the ECG signal. Segments of the data that are corrupted by MAs are removed, leading to some degree of data loss [6], as well as the increased labor associated with manual interpretation. It has previously been observed in one study that data collected in resting horses with textile electrodes had significantly lower percentages of MAs at approximately 35%, compared to data collected with Ag/AgCl electrodes at approximately 51% [21]. While a significant reduction in %MA was observed with textile electrodes in that study, over one-third of the data was still not of diagnostic quality and would therefore need to be excluded from all analysis. In comparison, we observed significantly lower %MA for both electrode types in the present study, despite identical experimental conditions of horses at rest, unrestrained in stalls, for one hour. In the present study, less than an average of 0.5% of the recording was found to be corrupted by motion for both electrode types, with no significant differences observed between the two.

Beyond the differences in electrode materials described above, there are many other possible explanations for the discrepancies observed between this study and data previously published using textile electrodes in horses. It has previously been shown that different yarn materials and manufacturing processes impact the skin-electrode contact impedance [41]. The contact impedance is in part dependent on skin preparation techniques, such as the application of saline [42,43]. It is possible that the conductive gel used in the present study outperformed the isopropyl alcohol used in previous equine studies [21,22]. It has been reported that increased electrode pressure is associated with reduced MAs [44,45], and it is possible that the integrated band utilized here was able to achieve more consistent pressure on the electrodes. Increased pressure is also associated with greater signal amplitude due to improved contact between the electrode and skin. While not explicitly stated, it appears that the peak signal amplitude recorded by Guidi et al. was approximately 0.5 mV, while our data recorded from the same locations showed an average peak amplitude greater than 1.5 mV. This may confirm that greater pressure was achieved by the current girth band design compared to the elastic band previously utilized. Interestingly, it has been found in human studies that increased electrode size (20–40 mm versus 5–10 mm) results in decreased skin-electrode impedance due to increased contact area [46,47]. We used much smaller textile electrodes (4 cm squares) than what has previously been used in horses (8 × 3 cm) [21,22], though this did not negatively impact signal amplitude. In addition to pressure and electrode size, it is possible that there is a variable and interactive effect of electrode location that impacts impedance. The electrodes used to calculate channel I in this study were most similar to the electrode locations used by Guidi et al. [21]. We calculated a mean peak amplitude of 1.87 ± 0.54 mV, indicating that despite the decreased electrode size, the pressure and locations utilized were able to achieve greater signal amplitudes. The possibly additive effects of skin preparation, band construction, and electrode location, size, and composition, clearly impact the ability of textile electrodes to maintain good contact with the skin, both at rest and during movement. Further investigation of other textile electrode materials and sizes is required to determine optimal combinations for equine ECG use. 

## 5. Conclusions

This study demonstrates that a smart textile girth band can be used to obtain very high-quality three-lead ECGs in horses at rest. Data quality did not differ between textile and Ag/AgCl electrodes, indicating that the textile electrodes composed of silver yarn are a practical and reliable alternative to the standard electrodes used. Further investigation into the utility and validity of textile electrodes in gathering ECG data during moderate- and high-intensity exercise is warranted. 

## Figures and Tables

**Figure 1 animals-12-03254-f001:**
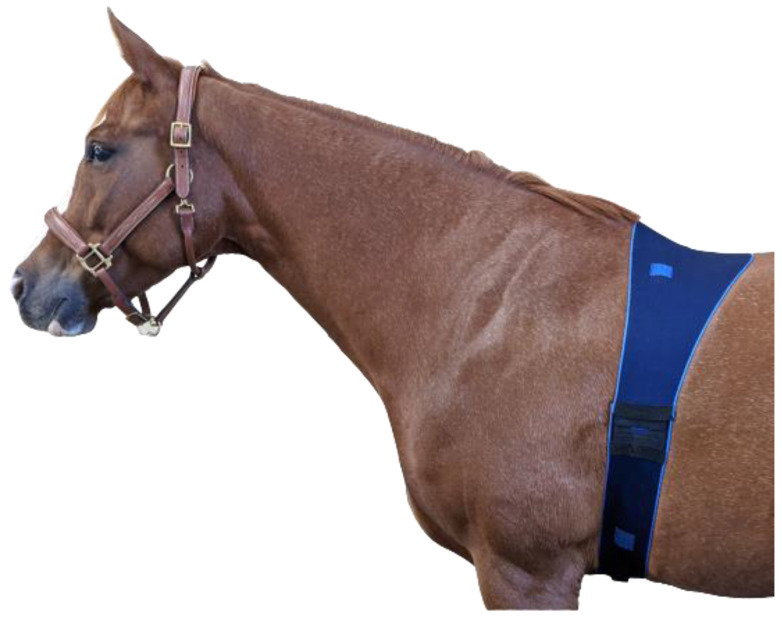
Girth band integrated with smart textile electrodes (light blue).

**Figure 2 animals-12-03254-f002:**
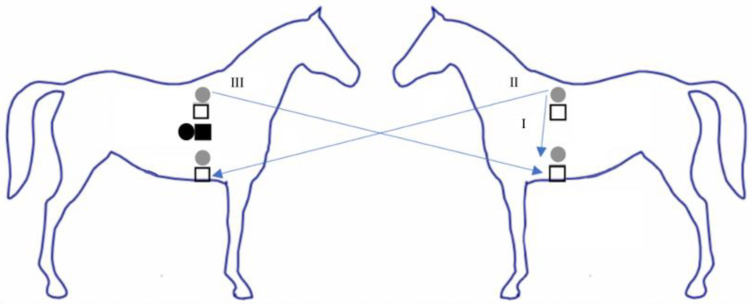
Electrode locations for textile electrodes (white squares) and Ag/AgCl electrodes (grey circles). The black square (textile) and black circle (Ag/AgCl) indicates the ground electrodes. Each of the three leads (I–III) is defined.

**Figure 3 animals-12-03254-f003:**
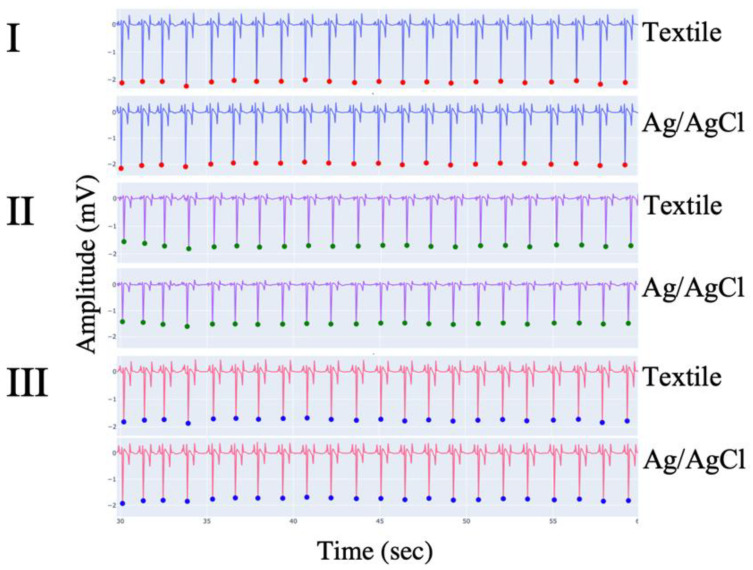
Example of ECG traces obtained simultaneously from the same horse using textile electrodes (**top** row) and Ag/AgCl electrodes (**bottom** row) for each of the three traces (I–III).

**Figure 4 animals-12-03254-f004:**
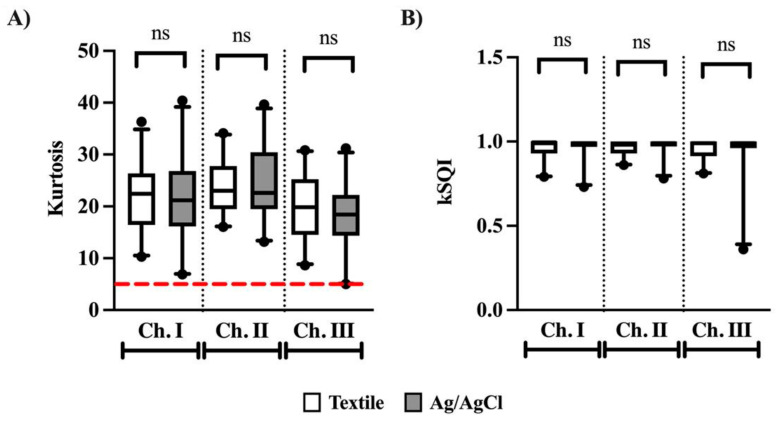
Kurtosis value (**A**) and kurtosis signal quality index kSQI, (**B**) calculated for textile (white) and Ag/AgCl (grey) electrodes for three channels. The red horizontal dotted line indicates a threshold of 5. Ns denotes no significance.

**Figure 5 animals-12-03254-f005:**
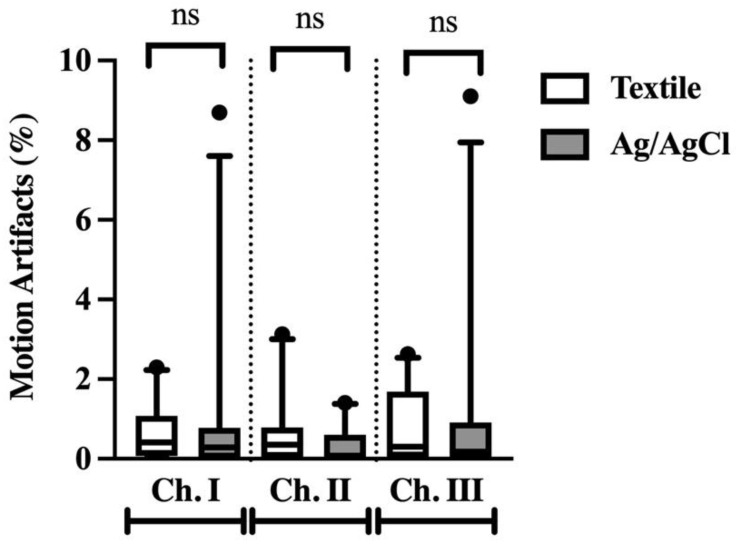
Percentage of motion artifacts for ECG data collected with textile (white) and Ag/AgCl (grey) electrodes. Ns denotes no significance.

**Figure 6 animals-12-03254-f006:**
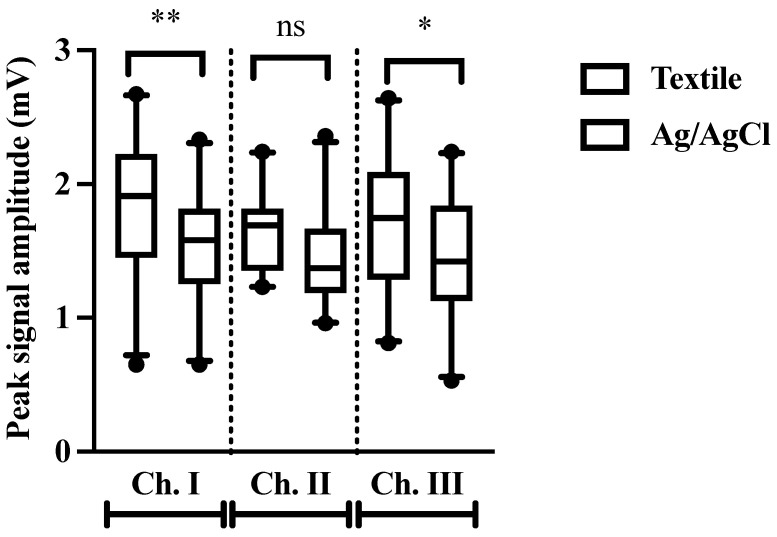
Peak signal amplitude measured in mV for ECG data collected with textile (white) and Ag/AgCl (grey) electrodes. Ns denotes no significance. * *p* < 0.05; ** *p* < 0.01.

**Figure 7 animals-12-03254-f007:**
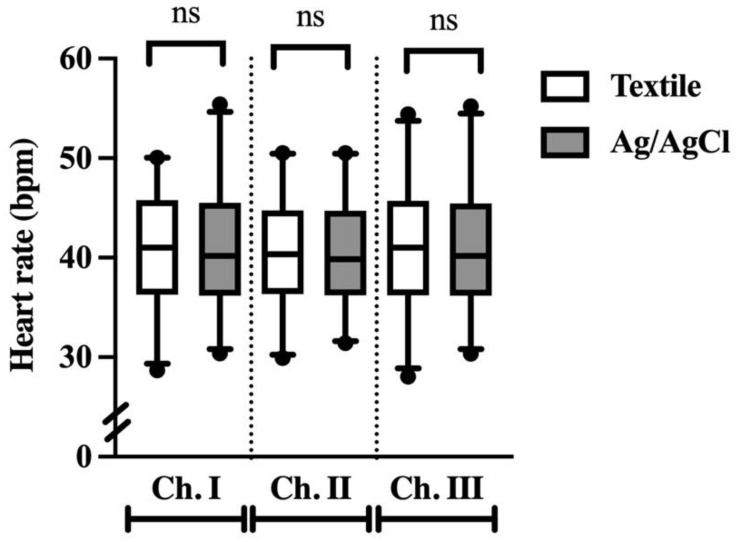
Heart rate output (beats per minute) for ECG data collected with textile (white) and Ag/AgCl (grey) electrodes. Ns denotes no significance.

## Data Availability

Data are available from authors upon reasonable request.

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
