# Peer review of "A Smart Textile Band Achieves High-Quality Electrocardiograms in Unrestrained Horses"

_animals, 2022, doi:10.3390/ani12233254_

Round 1

Reviewer 1 Report

This paper indeed demonstrates that smart textiles are a practical and reliable alternative to the standard electrodes typically used in ECG monitoring of horses. However, contrary to what the abstract suggests, this was already known, as has been acknowledged by the authors in lines 60-65. I urge the authors to be very clear in the abstract that they have a kind of ‘me-too-study’. They should also specify the difference between their smart textile electrodes and the ones used by Baragli’s group.

Regarding the methods:

-       Median filters can indeed remove baseline wander, but they introduce evident distortion in the detrended record. Quadratic variation reduction (QVR) has been shown to remove drift while not altering morphology of the complexes.

-       From the paper it’s not clear how the Kurtosis (k) value was derived. Instead of referring to a paper (line 120), authors should exactly describe how they estimated the Kurtosis (k) value, since this value is the core of their business. The paper by Li et al. Physiol Meas 2019 (section 2.1.3) cannot be simply applied cause it is unknown wat the discrete signal x-subscript-i is, whether you segmented the ECG, whether every sample within a window is assumed to have a Gaussian distribution, how you derived empirical estimates of the mean and standard deviation of x-subscript-i.

-       Mean baseline noise estimation (lines 128 – 131): what if the RR-interval is irregular? This will induce large differences between the maximum and mean of each sample across all the beats.

-       Mean baseline noise estimation (lines 128 – 131): The segment following the S-wave to the next P wave will include the T wave?

Regarding discussion:

-       Lines 213-215 are not clear

-       What are the implications of lines 233-234?

-       Why did Guidi et al. find such a high percentage of Mas (Lines 240-245)?

-       How was the contact between the band and the skin close to the whithers? From Figure 1 it appears to me that the textile electrodes are in danger of losing contact.

Reviewer 2 Report

I think it is a very interesting work. I am sure that the smart textile band will be attractive and very useful in the field of the equine cardiology. 

My comments on the article are brief:

Unless there are confidentiality constraints, I think the smart textile band could be described a little more, such as including the material it is made of or the measures.

L-256: you mention that you use a salt-free gel which has not been previously described in material and methods. I think this should also be reflected in this section.

L 329, quotation number 2, the year to be in bold type.

L 369 - 370: citation number 22, year to be in bold type.

Reviewer 3 Report

Interestingly, it provides practical knowledge on the use of ECG. The study does not show whether there was damage to the boade among tho horses? Have the horses been taught these straps or are they meeting for the first time? 

Round 2

Reviewer 1 Report

The changes made to the manuscript are much appreciated.

However, the description how kurtosis was estimated is still such that it cannot be repeated by others.

I suggest to change the text of lines 141 and further to:

ECG data was windowed into 10 sec segments, mutually overlapping by 5 sec. ECG quality of the segmented signals was compared using the Kurtosis (k) value, Kurtosis Signal Quality 141 Index (kSQI), percentage of motion artefacts (%MA), peak signal amplitude, mean 142 baseline noise, and heart rate (HR). K value is considered to be the single best metric for 143 assessing ECG signal quality, where the distribution of peaks is compared to a Gaussian 144 distribution [27]. For each segment and for each lead, the empirical estimate of kurtosis, , of the discrete signal xi of that lead was calculated by

where  and  are the empirical estimate of the mean and standard deviation of , respectively, and M is the number of samples [22,27]. As previously suggested, a k value greater than 5 indicates good quality ECG signal, while a k value lower than 5 indicates that the signal is impacted by motion artifacts or interference [22,27].

I also suggest to change lines 18 -19 to:

Smart textiles containing electrodes knit with stainless steel fibers have previously been shown to be a suitable alternative in horses at rest and during exercise. 

Please also note the typo in line 83: fibres -> fibers

Author Response

Thank you for your further comments. All changes requested by the reviewer have been made.